# Evaluation of Sociodemographic Factors and Prevalence of Oral Lesions in People Living with HIV from Cacoal, Rondônia, Amazon Region of Brazil

**DOI:** 10.3390/ijerph19052614

**Published:** 2022-02-24

**Authors:** Graziela de Carvalho Tavares da Rocha, Ricardo Roberto de Souza Fonseca, Aldemir Branco Oliveira-Filho, Andre Luis Ribeiro Ribeiro, Silvio Augusto Fernandes de Menezes, Rogério Valois Laurentino, Luiz Fernando Almeida Machado

**Affiliations:** 1Biology of Infectious and Parasitic Agents Post-Graduate Program, Federal University of Pará, Belem 66075-110, PA, Brazil; grazictr@hotmail.com (G.d.C.T.d.R.); ricardofonseca285@gmail.com (R.R.d.S.F.); valois@ufpa.br (R.V.L.); 2Virology Laboratory, Institute of Biological Sciences, Federal University of Pará, Belem 66075-110, PA, Brazil; 3Study and Research Group on Vulnerable Populations, Institute for Coastal Studies, Federal University of Pará, Bragança 68600-000, PA, Brazil; olivfilho@ufpa.br; 4Private Dental Practice, INCOM—Instituto de Cirurgia Oral e Maxilofacial, Belem 66050-350, PA, Brazil; andre.ribeiro.13@ucl.ac.uk; 5Department of Periodontology, University Center of State of Pará, Belem 66060-575, PA, Brazil; menezesperio@gmail.com

**Keywords:** dental care, epidemiology, HIV, public health, oral health

## Abstract

Background: It is necessary to evaluate and understand the prevalence and risk factors of oral lesions (OL) in people living with HIV (PLWH) who were never studied before. The present study aimed to describe the prevalence of OL and its correlation with CD4^+^ T lymphocytes counts and HIV plasma viral load in PLWH treated in Rondônia. Methods: A cross-sectional study was carried out at Cacoal city, Rondônia state, Northern Brazil. Sociodemographic, epidemiological, immunologic and virological information of 113 PLWH were collected from medical records and dental examination was conducted to diagnose and classify OL in PLWH. Statistical analysis was performed using relative frequency distribution, ANOVA, Kruskal–Wallis, T-student and Mann-Whitney tests. Results: The overall prevalence of oral lesions was 28.3% (32/113), with candidiasis (7/32; 21.8%) and aphthous ulcer (7/32; 21.8%) being the most prevalent. There was a predominance of females, most patients being married, with a low level of education, a family income of 1 to 3 minimum wages and a single partner. An association was observed between the presence of oral lesions and a high viral load, as well as a lower occurrence of oral lesions in individuals with a higher count of CD4^+^ T cells. Conclusions: This study reveals a low prevalence of OL among PLWH, as well as the absence of relationship between HIV viral load, CD4^+^ T cells count and OL high prevalence.

## 1. Introduction

According to data from the Joint United Nations Program on HIV/AIDS (UNAIDS), it is estimated that there are about 2.1 million people living with HIV in Latin America and the Caribbean in 2019 [1]. In Brazil, UNAIDS stipulated that in 2019, approximately 920,000 people are living with HIV, 48,000 new infections in 2019 and 14,000 people died due to complications from HIV [2]. According to the Brazilian Ministry of Health, 41,909 new cases of HIV, 37,308 cases of Acquired immunodeficiency syndrome (AIDS) were diagnosed and a detection rate of 17.8/100 thousand inhabitants [3].

The Northern region of Brazil has the second lowest rate of HIV infection reported, with 25,966 (7.6%) reported cases between 2007 and June 2020, in 2020 the state of Rondônia had the third highest rate in confirmed cases of HIV with 2331, in confirmed cases of AIDS with 6576 and fifth in the detection rate of AIDS cases with 17.8/100 thousand, making Rondônia the twelfth federation in Brazil in cases of HIV [3]. In addition, in the northern region of Brazil, some studies have documented the prevalence of HIV and its oral manifestations; it is known that HIV affects the individual’s immune system, triggering various disorders, neoplasms and opportunistic infections, such as lesions in the oral cavity [4,5,6].

Oral lesions (OL) are initial and common clinical features in people living with HIV (PLWH). Oral manifestations are usually accurate indicators of immunosuppression and can also be used as a means of presumptive diagnosis of HIV infection [7,8,9,10]. There are about 24 types of OL in the literature linked to HIV and can be classified as fungal, viral and bacterial infections, or neoplasms such as Kaposi’s sarcoma, nonspecific aphthous ulcerations and salivary gland diseases [11,12]. Even with the decline in the occurrence of OL due to the use of antiretroviral therapy (ART), the most reported OL worldwide are: oral candidiasis, hairy leukoplakia, Kaposi’s sarcoma, nonspecific ulcerations and periodontal disease [13,14,15,16].

OL significantly contributes to the morbidity of PLWH, affecting the functionality of the oral cavity, food, speech, aesthetics and even social interactions and may affect the individual’s psychological state, so the quick and accurate diagnosis of the patient is essential for infection control for HIV and AIDS prevention [5,6,17,18]. Although the oral manifestations associated with HIV are well documented in the literature [19,20], to date there is a lack of information about HIV infection and its oral manifestations in the northern region of Brazil. In two studies were carried out in the state of Pará, where in one the occurrence of LO was 47%, with candidiasis (28%), periodontal disease (28%) and cervical-facial lymphadenopathy (17.5%) being the most common and in another, the most prevalent OL in the PLWH population were caries (32.6%), candidiasis (32%) and periodontal disease (17%) [4,5,6]. This study describes, for the first time, the prevalence of oral lesions in a group of PLWH assisted in the city of Cacoal, state of Rondônia, northern region of Brazil and its correlation with HIV plasma viral load levels and the number of CD4^+^ T lymphocytes.

## 2. Materials and Methods

### 2.1. Study Design and Area Knowledge

Northern Brazil is home to the tropical region of the Amazon which, in this country, is a rural, socioeconomically underdeveloped region with high levels of poverty, limited transport infrastructure and absent or inadequate health services or difficult access due to watersheds, distances territorial long and mismatched public roads. As dental services are one of the most affected by geographic and socioeconomic conditions in the region, as well as the difficulty of accessing information and low education levels, these can lead to the proliferation of dental caries and periodontal disease (PD).

This descriptive cross-sectional study was based on biological and socioeconomic data from samples of PLWH attended at the outpatient clinic for sexually transmitted infections (STIs), from the Specialized Care Service (SCS), provided by the Specialized Unit National Health Foundation (FNS) located in the municipality of Cacoal, Rondônia. The municipality of Cacoal has an estimated population of 85,359 inhabitants, with an average monthly salary of 2 minimum wages, per capita GDP R$ 25,708.9620, but it is also a regional center in oral health, very important for smaller municipalities in the region, providing care to all of them systematically, including the PLWH (Figure 1).

### 2.2. Ethics 

The project was submitted and approved by the Ethics and Research Committee of the Faculty of Biomedical Sciences of Cacoal—FACIMED under protocol number 1015-13.

### 2.3. Clinical Parameters

The clinical parameters were established according to the Classification of HIV-related oral lesions (Shiboski et al. 2009 and Menezes et al. 2015). The oral conditions were divided by their etiology and type of oral lesion and are presented as 1—Fungal Infections: Pseudomembranous candidiasis; Erythematous candidiasis and Angular cheilitis/2—Viral Infections: Hairy leukoplakia; Oral wart; Herpes labialis and Intraoral herpes simplex/3—Idiopathic Conditions: Aphthous stomatitis, Ulceration not otherwise specified and Necrotizing stomatitis/4—Bacterial Infections: Necrotizing gingivitis or periodontitis/5—Salivary gland diseases: Parotid enlargement and Salivary hypofunction/6—Neoplasms: Oral Kaposi sarcoma, Oral non-Hodgkin’s lymphoma and Oral squamous cell carcinoma.

### 2.4. Collection of Samples and Personal Data

The sample consisted of patients registered and treated at the SCS in Cacoal, Rondônia. From January 1999 to January 2013, 161 individuals were informed about the purpose of the study and invited to participate; only 113 agreed and signed written consent form before data collection and dental evaluation.

The study eligibility criteria were: (i) individuals with a confirmed diagnosis of HIV-1 infection; (ii) registered serviced at the SCS in Cacoal, Rondônia; (iii) at least 20 teeth in the mouth and without periodontal therapy for about 1 year; (iv) medical records duly filled in; (v) regular users of ART; (vi) monthly follow-up at the SCS in Cacoal; (vii) oral lesions described by Menezes et al. (2015) present in Table 1. The exclusion criteria were: (i) individuals who were transferred to other locations; (ii) Individuals who started follow-up at the SCS but no longer returned to monthly follow-up; (iii) medical records not filled out correctly and (iv) users of complete dentures.

Each participant was physically and orally evaluated in a private location in the SCS. Clinical data were collected by a single researcher, specialist in oral pathology, previously calibrated by Kappa test and with previous experience in clinical studies. The intraoral clinical examination was performed in a dental office, dental chair, under indirect and artificial light, using a dental mirror, Williams periodontal probe (Hu-Friedy, Chicago, IL, USA) and clinical tweezers, all sterile, consisting of disposable materials; the evaluations were performed monthly between January 1999 and January 2013 to maximum analysis of patients. To differ OL and improve diagnosis, while evaluating if there is a doubt regarding the OL or manifestations seemed a more advanced disease, a histopathologic examination was performed to define the diagnosis.

After the intraoral clinical examination, the collection of sociodemographic data, HIV plasma viral load and CD4^+^ T lymphocyte count (LTCD4+) occurred from information available in the medical records, seeking the information obtained at the closest date of the intraoral clinical examination.

### 2.5. Statistical Analysis

All statistical procedures were performed using SPSS 21.0 for Windows (SPSS Inc., Chicago, IL, USA). A descriptive analysis of the data was performed, with distribution of relative frequencies, using the minimum value, maximum value and median for continuous quantitative variables, and then the data were categorized and grouped. To analyze the association between oral parameters with time of HIV infection, LTCD4+ count and HIV plasma viral load, ANOVA, Kruskal–Wallis, T-student and Mann–Whitney tests were used, with a reliability index of 95% (CI = 95%) and the level of statistical significance adopted was 5% (*p* < 0.05).

## 3. Results

### 3.1. Sample 

In total, 161 PLWH were recruited for the study evaluation: all were residents of the city of Cacoal and its surroundings, but 43 were from Cacoal and 70 were from other regions of Brazil, and of these, half were from the South region (35/70; 50.0%), followed by the Southeast (22/70; 31.4%), Midwest (8/70; 11.4%) and Northeast (5/70; 7.1%) regions. However, 48 of them were excluded from the study (19 did not sign the consent form, 15 changed cities and 14 did not meet the clinical criteria); thus, the final sample of this study was 113 PLWH.

### 3.2. Social, Behavioral and Health Characteristics of PLWH

The mean age was 39.7 (±13.1 years) and the age group with the highest number of cases was between 36 and 52 years with 53 individuals (46.9%). Most study participants were female (59/113; 52.2%), residents of urban areas (96/113; 85.0%), self-declared white or brown (both with 52/113; 46%), married (52/113; 46%), heterosexual (95/113; 84.1%), were illiterate or had incomplete primary education (43/113; 38%) and as for the family income, most obtain 1 at 3 minimum wages (66/113; 58.4%) per month.

As for the risk factors for HIV infection, most individuals (except the children participating in the study) declared that they had had, in the last 30 days, only one sexual partner (56/113; 49.6%), sometimes having sex safe (52/113; 46%), used illicit drugs (59/113; 52.2%) and regarding tattoos, most reported not having (89/113; 78.8%), as well as having no history of STIs (75/113; 66.3%), with all socioeconomic, demographic and behavioral data (Table 1).

### 3.3. Oral Manifestations Prevalence 

Among all 113 individuals evaluated by the study, only 32 (28.7%) had positive results for the presence of lesions in the oral cavity. In the sample of 32 PLWH with oral manifestations related to HIV, on a decreasing prevalence scale, we cite: candidiasis (7/32; 21.8%) and aphthous ulcer (7/32; 21.8%) were the most prevalent OL followed by herpes simplex (6/32; 18.75%), herpes zoster (5/32; 15.6%), hairy leukoplakia (5/32; 15.6%), linear gingival erythema (1/32; 3.12%) and grade 1 periodontitis (1/32; 3.12%). Of the 32 PLWH, 28 (87.5%) had only 1 lesion and 4 (12.5%) had 2 lesions, among the places where the lesions were present, most were on the dorsum of the tongue, palate, lips, buccal mucosa and attached gingiva (Table 2).

Among the 32 PLWH who presented some alteration in the oral cavity, 19 (59.4%) were male and 13 (40.6%) were female, without statistical significance (*p* = 0.122). With regard to age, the most prevalent age group was 36 to 52 years old (16; 50.0%), with a statistically significant difference (*p* = 0.024) in relation to other age groups. As for the analysis of sexual orientation, condom use and number of partners, there was no statistically significant result to be associated with oral manifestations (Table 3).

### 3.4. Quantification of LTCD_4_^+^ and HIV-1 Plasmatic Viral Load

Regarding the HIV plasma viral load, 31.9% (36/113) of the patients had undetectable values (≤50 copies/mL), 22.1% (25/113) had a viral load lower than 10,000 copies/mL, 31.9% (36/113) had numbers between 10,000 and 100,000 copies/mL, 7.1% patients (8/113) had a viral load above 100,000 copies/mL and for 6.3% (7/113) it was not possible to obtain this information.

As for the LTCD4+ count of the 106 PLWH, it was found that the mean was 475 ± 239 cells per mm3 of blood, and most patients (61/106; 54%) exhibited between 200 and 500 cells, 39% (44/106) had levels above 500 cells/mm3 of blood and only 0.9% of the patient (1/106) had a count below 50 cells/mm3.

Regarding the presence of lesions in the oral cavity and the HIV plasma viral load levels presented by these patients, it was possible to notice that the increase in viral load levels significantly increases the number of cases of oral lesions (*p* = 0.043; Mann–Whitney test). Therefore, it was found that the higher the number of LTCD4+ per mm3, the lower the occurrence of lesions in the oral cavity of these patients (*p* = 0.045; Mann—Whitney test).

## 4. Discussion

The present study evaluated the oral situation of patients and the prevalence of oral pathologies resulting from immunosuppression caused by HIV, as well as the associated sociobehavioral and health risk characteristics in a sample of PLWH in the city of Cacoal-RO northern Brazil, making this the first epidemiological report focused on this vulnerable population in a remote region of Brazil and never before studied. Furthermore, the need for the study arose due to the fact that at the time of the clinical evaluations, the state of Rondônia had only six specialized services for the care of PLWH, two located in the capital Porto Velho and only one in the city of Cacoal, establishing the municipality as a regional health center, PLWH for satellite municipalities such as Espigão D’Oeste, Minister Andreazza, Pimenta Bueno, Primavera de Rondônia and São Felipe.

In addition to identifying the viral load, LTCD4+ count and risk factors, we sought to correlate the presence, quantity and location of the OL of the individuals evaluated and we observed a significant association between the HIV viral load and the presence of OL, that is, when elevated to viral load had LO, such as oral candidiasis in individuals [10,13]. Inversely proportional to the previously presented data, a significant association can be observed between the decay of the LTCD4+ count and the presence of LO. In order to corroborate the data found in the literature, as well as a national and global trend, the data from this study showed that when there is an increase in viral load and lymphocyte decline, there may be intraoral clinical data that may provide a possible prognosis of the patient’s systemic health and probability of developing AIDS [14,16].

The data, related to the prevalence of *Candida albicans* in the oral cavity, presented by this study corroborate the data evidenced by the study by Lourenço et al. [14] with a population living with HIV, but in the city of Ribeirão Preto, southeastern Brazil. According to the authors, a greater amount of *Candida spp* strains could be found in the oral cavity of patients as viral loads increased and lymphocyte counts decreased; however, the authors concluded that PD is a relevant factor for manifestations candidiasis clinics and development of a co-infection, which differs from our study because PD had a low prevalence.

In an attempt to explain the reason for the high prevalence of candidiasis among OL, according to Heron and Elahi [21] important cells of the immune system such as Th17 cells and Interleukin 22 (IL-22) can mediate cellular immunity and act on the integrity of the immune barrier against pathogens such as HIV or *C. albicans* on the surfaces of the oral mucosa, so these cells will stimulate the epithelial cells to produce antimicrobial factors to eliminate fungi and bacteria, promoting inflammation through the induction of inflammatory cytokines, chemokines and recruitment of neutrophils, thus, in addition to immunosuppression due to LTCD4+ depletion, it is possible to infer that the involvement of Th17 and IL-22 cells may be associated with a greater susceptibility to oral candidiasis [21,22]. Yet, according to the authors, another way to explain the ease of HIV-positive individuals to develop conditions such as candidiasis is due to the low production of nitric oxide (NO) produced by endothelia, epithelia and macrophages of oral tissues; in the study it is indicated that the production of reactive oxygen species and nitrogen radicals, such as NO, can deteriorate or even inhibit the growth of C. albicans in the oral cavity. However, PLWH has presented strains that produce flavohemoglobin genes which are resistant to NO [21,22].

Interestingly, a fact that caught the attention of the authors of this article was the low incidence of PD among the individuals evaluated; this data was not expected mainly because the absence of periodontal therapies in a period of 12 months was among the inclusion criteria of the study. PD is currently characterized as an infectious-inflammatory disease that can be chronic or acute and directly affects the dental support tissues, and its main cause is linked to the accumulation of dental biofilm, formation of gingival calculi and initial colonization by aerobic microorganisms gram positive and later adhesion of gram negative anaerobic bacteria which triggered local pro-inflammatory processes in periodontal soft tissues, dysbiotic state, increased osteoclastic activity resulting in loss of clinical attachment levels and, in severe cases, tooth loss [4,5,6].

In contrast to the results presented for PD in this study, Bodhade et al. [22] evaluated 399 PLWH and observed that necrotizing periodontal diseases (gingivitis and necrotizing periodontitis) was the second most prevalent OL among patients with 55 (13.8%) being second only to oral candidiasis; it was the most prevalent OL with 157 (39.3%) of the individuals evaluated. In another study located in the city of Belém, northern Brazil and eastern Amazon, 79 PLWH were analyzed and, similarly to the study by Bodhade et al. [22], PD was the second most prevalent OL with about 28% of affected patients; in other words, contradicting our data, PD is among the most frequent oral manifestation in PLWH due to the severity and progression of PD being determined by factors related to the host’s response, in addition to the presence and virulence of bacteria and by compromising the main cells of the immune system, as the HIV progressively alters the body’s immune response, favoring the formation of more active periodontal pockets and more aggressive microbiota.

The socioeconomic characteristics of the PLWH in this study were consistent with information reported by other studies also carried out in northern Brazil—adults, married, heterosexual, with low educational level and income, who practice unprotected sex. This study also pointed out some specific behavioral risk characteristics, such as the vast diversity of sexual partners in a given period, use of illicit drugs, use of condoms, presence of tattoos and history of STIs—a single sexual partner, sometimes practiced safe sex, used illegal drugs and did not have tattoos as well as a history of STIs. The data collected in this study show similar characteristics of PLWH that were also reported in the city Belém, capital of Pará, demonstrating a possible trend in the northern region, so this information related to risky behavioral characteristics in northern Brazil is important and should be investigated to inform prevention and treatment measures targeted at this population by the federal and state governments [4,5,6].

Another relevant point that might have directly influenced the OL prevalence in PLWH in this study was the regular use of antiretroviral therapy (ART). Since the advent of ART, the life expectancy of PLWH increased, as well as access to information through the internet, and then so too did the demand for dental treatment, including oral rehabilitation with dental implants, which improved dental quality of life significantly in the last few years [23,24,25]. The attempt to rehabilitate PLWH with dental implants has been reported since 1998 [26], and since then papers seems to focus on healing, osseointegration process, longevity of dental implants, guided bone regeneration and peri-implant diseases. In most studies, the success of dental implant therapy in PLWH can be observed in the most diverse clinical situations and commonly in this study we found that the use of ART and medical control of comorbidities will directly influence the success of rehabilitation and the quality of life of the patient [27,28].

This pioneering study on the oral health of PLWH in an unprecedented city in the North of Brazil naturally has certain limitations. First, it should be noted that this study focused on a specific population and had a refusal to participate rate of approximately 30%. Refusal to volunteer to participate in this research, presumably due to the unfortunate stigma that still surrounds HIV infection, is an issue. Another issue for refusal reported by individuals was that no clinical and scientific data was returned to patients, causing them to discourage voluntary participation. There are also difficulties of locomotion for patients from municipalities neighboring the SCS in Cacoal. Other limitations are related to self-reported information, such as consumption of illicit drugs, sexual orientation, number of sexual partners, history of STIs, ART use, tobacco and alcohol use and other personal and sensitive information for the participants; this missing information may compromise the complete understanding of the presented data and possible improvements in the care of patients treated in the SOE.

## 5. Conclusions

Despite the low prevalence of oral lesions detected in PLWH at Cacoal, this paper highlighted the importance of dental treatment and management in key populations like PLWH in northern Brazil, which historically is a region with difficulties in key populations accessing quality dental services. The study also demonstrated that oral candidiasis and aphthous ulceration were the most prevalent OL and those lesions could be associated to risky sexual behavior, absence of dental management, low LTCD4+ count and illicit drug use, which were relevant factors for the appearance of OL. Therefore, we concluded that key populations such as PLWH at Cacoal city need urgently easier access to dental treatment/management to prevent any possible OL and to continue the correct management of HIV.

## Figures and Tables

**Figure 1 ijerph-19-02614-f001:**
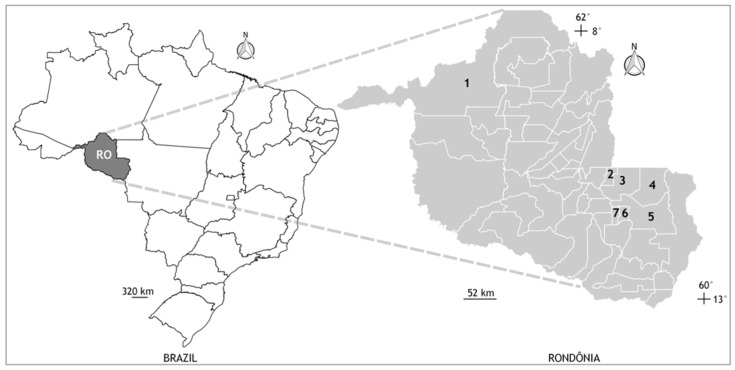
Geographic location of the municipalities of Porto Velho (state capital) and Cacoal (site of the National Health Foundation sexually transmitted infections clinic), state of Rondônia (RO), northern Brazil. All sample and personal data collections from the participants were carried out in the city of Cacoal, but some participants lived in other nearby locations. Municipalities: (1) Porto Velho, (2) Cacoal, (3) Espigão D’Oeste, (4) Ministro Andreazza, (5) Pimenta Bueno, (6) Primavera de Rondônia, and (7) São Felipe.

**Table 1 ijerph-19-02614-t001:** Socioeconomic profile and risk behavior related to HIV infection among people with HIV/AIDS.

Parameters	HIV	
	*n*	%
Age (Years)		
4–20	4	3.5
20–36	36	31.9
36–52	53	46.9
52–68	16	14.2
68–84	4	3.5
Gender		
Male	54	47.8
Female	59	52.2
Colour/race (self-identified)		
White	52	46
Black	52	46
Pardo (mixed race)	8	7.1
Indigenous	1	0.9
Marital status *		
Single	37	32.7
Married	52	46
Separated	10	8.8
Widowed	6	5.3
Not informed	8	7.1
Monthly income (Brazilian minimum wage) *		
Up to one wage	14	12.4
1–3 wages	66	58.4
4–6 wages	15	13.3
7–11 wages	3	2.6
Not informed	15	13.3
Sexual orientation		
Homosexual	12	10.7
Heterosexual	95	84.1
Bisexual	3	2.6
Not informed	3	2.6
Source		
Capital city	96	85
Countryside	17	15
Length of education *		
No formal education (including illiterates)	43	38
Up to elementary school	27	23.9
Up to high school	29	25.7
College graduation	8	7.1
Not informed	6	5.3
Condom use **		
Rarely	7	6.2
Never	18	15.9
Sometimes	52	46
Always	36	31.9
Number of sexual partners **		
0–1	56	49.6
2–5	46	40.7
Up to 5	11	9.7
STI History		
Yes	38	33.7
No	75	66.3
Illicit drug use **		
Yes	59	52.3
No	54	47.7
Alcohol use **		
Yes	38	33.7
No	75	66.3
Tobacco use **		
Yes	50	44.2
No	63	55.7
ART use *		
Yes	90	79.6
No	23	20.4
Tattoos		
Yes	24	21.2
No	89	78.8

* Over last 12 months, ** Over last 30 days.

**Table 2 ijerph-19-02614-t002:** Frequency of oral lesion type per group.

Parameters	Oral Lesion Types (*n*/%)					
	Fungal Infections	Viral Infections	Idiopathic Conditions	Bacterial Infections	Salivary Gland Diseases	Neoplasms
Anatomic region						
Maxilla	-	2 (12.5%) ^‡^	-	1 (50%) ^¶^	-	-
Mandible	-	3 (18.75%) ^‡^	-	1 (50%) ^#^	-	-
Tongue	4 (57.1%) *	4 (25%) ^†^	1 (14.3%) **	-	-	-
Lips and buccal mucosa	3 (42.9%) *	5 (31.25%) ^¥^	6 (85.7%) **	-	-	-
Oropharynx	-	2 (12.5%) ^†^	-	-	-	-
Salivary glands	-	-	-	-	-	-

* candidiasis/** aphthous ulcer/^†^ herpes simplex/^‡^ herpes zoster/^¥^ hairy leukoplakia/^¶^ linear gingival erythema/^#^ periodontitis.

**Table 3 ijerph-19-02614-t003:** Association between oral lesions prevalence, sociodemographic profile and risk factors in patients living with HIV/AIDS treated at the Specialized Outpatient Service (SOE) from January 1999 to January 2013.

Parameters	Oral Manifestations					
	Yes			No		*p*
	*n*	%		*n*	%	
Gender						
Female	13	40.6		46	56.8	0.122
Male	19	59.4		35	43.2	
Total	32	100.0		81	100.0	
Age						
4–20	3	9.4		1	1.2	0.024
20–36	6	18.8		35	43.2	
36–52	16	50.0		32	39.5	
52–68	7	21.9		10	12.3	
68–84	0	0.0		3	3.7	
Total	32	100.0		81	100.0	
Sexual orientation						
Homosexual	0	0		3	3.7	0.298
Heterosexual	25	78.1		70	86.4	
Bisexual	5	15.6		7	8.6	
Not informed	2	6.3		1	1.2	
Total	32	100.0		81	100.0	
Condom use						
Rarely	4	12.5		3	3.7	0.347
Never	4	12.5		14	17.3	
Sometimes	17	53.1		35	43.2	
Always	7	21.9		29	35.8	
Total	32	100.0		81	100.0	
Number of sexual partners						
0–1	16	50		40	49.4	0.853
2–5	13	40.6		33	40.7	
Up to 5	3	9.4		8	9.9	
Total	32	100.0		81	100.0	

## Data Availability

The data that support the findings of this study are available from the corresponding author, L.F.A.M., upon reasonable request.

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
