# Peer review of "Evaluation of Sociodemographic Factors and Prevalence of Oral Lesions in People Living with HIV from Cacoal, Rondônia, Amazon Region of Brazil"

_ijerph, 2022, doi:10.3390/ijerph19052614_

Round 1

Reviewer 1 Report

The design and development of the research seems to me to be correct in order to be able to respond to the proposed objectives:

1 To know the prevalence of oral lesions in a group of people living with HIV/AIDS assisted in the city of Cacoal, state of Rondônia, northern region of Brazil.  For this purpose a cross-sectional descriptive study was carried out.

2 To analyze the relationship of oral lesions with socioeconomic, demographic and behavioral variables as well as their relationship with HIV viral load and CD4+ T cells count.  For this purpose, an analytical study was carried out.

Some minor comments may help to make the work more attractive to the reader:

1, What was the sample recruitment process? How were the 161 individuals recruited? were recruited consecutively, over a period of time, randomly from the total population?

2, I don't understand properly if each patient is screened for 3 months.

3, Even if they are very well known initials, the first time STIs appear, they should be defined.

I think you should revise the conclusions and focus exclusively on the results of their study.

Author Response

Reply to reviewer #1

1. Concern of the reviewer

• What was the sample recruitment process? How were the 161 individuals recruited? were recruited consecutively, over a period of time, randomly from the total population? 

Our response: Dear Reviewer #1, we appreciate your suggestion and the text was carefully revised.  

Revised text: Page 3, lines 121-123, “From January 1999 to January 2013, 161 individuals were informed about the purpose of the study and invited to participate, only 113 agreed and signed written consent form before data collection and dental evaluation.” 

2. Concern of the reviewer

• I don't understand properly if each patient is screened for 3 months. 

Our response: Dear Reviewer #1, we appreciate your concern, all patients were evaluated once during their monthly HIV treatment, the rest of the data were collected retrospectively in medical and dental files. 

3. Concern of the reviewer

• Even if they are very well known initials, the first time STIs appear, they should be defined. 

Our response: Dear Reviewer #1, we appreciate your suggestion and the text was carefully revised.  

Revised text: Page 1, line 42, “Acquired immunodeficiency syndrome (AIDS)”. 

4. Concern of the reviewer

• I think you should revise the conclusions and focus exclusively on the results of their study. 

Our response: Dear Reviewer #1, we appreciate your suggestion and the text was carefully revised.  

Revised text: Page 8, lines 360-369, “Despite the low prevalence of oral lesions detected in PLWH at Cacoal, this paper highlighted the importance of dental treatment and management in key populations like PLWH in northern Brazil, which historically is a region that has difficulties to key populations to access quality dental services. The study also demonstrated that oral candidiasis and aphthous ulceration were the most prevalent OL and those lesions could be associated to risky sexual behavior, absence of dental management, low LTCD4+ count and illicit drugs use were relevant factors for the appearance of OL. Therefore, we concluded that key population as PLWH at Cacoal city needs urgently an easier access to dental treatment/management to prevent any possible OL and to continue the correct management of HIV.”

Reviewer 2 Report

Dear authors,

the Manuscript has merit,

although there are some points that need to be reviewed.

Page 1, line 21: << Please correct UNDERSTAND>>.

Page 1, line 34: << Please correct STUDY >>.

Page 8, Discussion.

Please add a brief paragraph to make Discussion more interesting to the reader. HIV investigation in a population could be very usefull to get to know about feasibility of dental implant treatment is that specific  population. In fact, like other systemic diseases such as diabetes mellitus, HIV might be a risk factor for dental implants survival rate. Particularly, presence of CD4+ T cells was demonstrated play a role also on fixtures osseointegration.

Please cite PMID 33412779

Please cite PMID 26238779

PMID 32694022
PMID 34496113
PMID 26238779
PMID 33121608
PMID 33412779

Author Response

Reply to reviewer #2

1. Concern of the reviewer             

• Page 1, line 21: << Please correct UNDERSTAND>>. 

Our response: Dear Reviewer #2, we appreciate your concern. The text was carefully added. 

Revised text:Page 1, line 20, “understand.” 

2. Concern of the reviewer

• “Page 1, line 34: << Please correct STUDY >>.” 

Our response: Dear Reviewer #2, we appreciate your comment.

 Revised text: Page 1, line 34, “study.” 

3. Concern of the reviewer

  • Please add a brief paragraph to make Discussion more interesting to the reader. HIV investigation in a population could be very usefull to get to know about feasibility of dental implant treatment is that specific  population. In fact, like other systemic diseases such as diabetes mellitus, HIV might be a risk factor for dental implants survival rate. Particularly, presence of CD4+ T cells was demonstrated play a role also on fixtures osseointegration.

Our response: Dear Reviewer #2, we appreciate your suggestion and concern. The text was carefully added. 

Revised text: Page 8, lines 342-353,Another relevant point that might directly influenced the OL prevalence in PLWH in this study is the regular use of antiretroviral therapy (ART), since the advent of ART, the life expectancy of PLWH increased, as well as the access to information through internet, so the demand for dental treatment, including oral rehabilitation with dental implants, to improve dental quality of life increased significantly in the last few years [23-25]. The attempt to rehabilitate PLWH with dental implants has been reported since 1998 [26], since then papers seems to focus on healing, osseointegration process, longevity of dental implants, guided bone regeneration and peri-implant diseases. In most studies, the success of dental implant therapy in PLWH can be observed in the most diverse clinical situations and commonly in this study we found that the use of ART and medical control of comorbidities will directly influence the success of rehabilitation and the quality of life of the patient [27,28].” 

Reviewer 3 Report

Study about Sociodemographic factors and the oral lesions in HIV-positive patients is interesting. However, the manuscript needs revision as follows:

  1. According to the results, it is better to be modified the title as follows: “Evaluation of Socio-demographic factors and prevalence of oral lesions in people living with HIV from Cacoal, Rondônia, Amazon Region of Brazil.
  2. According to the title and the aim of the study, it is necessary to prepare a table that summarizes the data about oral lesions including the frequency of every six groups and also the type of lesions in the groups with number and percentage. Also, the number and percentage of different intraoral sites involvement should be mentioned in the table.
  3. As mentioned in lines 110 to 119, oral conditions were divided into 6 types clinically. However, neoplasms only can be diagnosed with histopathologic examination. The question is how the authors “ruled out/in” the neoplasms such as Kaposi sarcoma, non-Hodgkin’s lymphoma, and oral squamous cell carcinoma without histopathologic examination.

Author Response

Reply to reviewer #3

1. Concern of the reviewer             

• According to the results, it is better to be modified the title as follows: “Evaluation of Socio-demographic factors and prevalence of oral lesions in people living with HIV from Cacoal, Rondônia, Amazon Region of Brazil. 

Our response: Dear Reviewer #3, we appreciate your suggestion. The text was carefully added. 

Revised text:Page 1, lines 1-3,Evaluation of Socio-demographic factors and prevalence of oral lesions in people living with HIV from Cacoal, Rondônia, Amazon Region of Brazil” 

2. Concern of the reviewer             

• According to the title and the aim of the study, it is necessary to prepare a table that summarizes the data about oral lesions including the frequency of every six groups and also the type of lesions in the groups with number and percentage. Also, the number and percentage of different intraoral sites involvement should be mentioned in the table. 

Our response: Dear Reviewer #3, we appreciate your concern. The table was carefully added. 

3. Concern of the reviewer             

• As mentioned in lines 110 to 119, oral conditions were divided into 6 types clinically. However, neoplasms only can be diagnosed with histopathologic examination. The question is how the authors “ruled out/in” the neoplasms such as Kaposi sarcoma, non-Hodgkin’s lymphoma, and oral squamous cell carcinoma without histopathologic examination. 

Our response: Dear Reviewer #3, we appreciate your concern. We know about the histopathologic examination necessity in neoplasms cases; however, a clinical examination was performed at first because extra and intraoral clinical examination must be the first step towards diagnosis, which we could use histopathologic examination as a resource in our diagnosis. So, if clinical symptoms suggested a neoplasms or oral lesion manifestations seemed a more advanced disease the dentist previous calibrated and experient in clinical studies asked the histopathologic examination. 

Revised text: Page 3, lines 133-142,Each participant was physical and oral evaluated in a private location in the SCS. Clinical data were collected by a single researcher, specialist in oral pathology, previously calibrated by Kappa test and with previous experience in clinical studies. The intraoral clinical examination was performed in a dental office, dental chair, under indirect and artificial light, using a dental mirror, Williams periodontal probe (Hu-Friedy, Chicago, IL, USA) and clinical tweezers, all sterile, consisting of disposable materials, the evaluations were performed monthly during January 1999 to January 2013 to maximum analysis of patients. To differ OL and improve diagnosis, while evaluating if there is a doubt regarding the OL or manifestations seemed a more advanced disease a histopathologic examination were performed to define the diagnosis.

Reviewer 4 Report

I appreciate the attempt to picture local epidemiology of oral lesions in people with HIV in a remote area of Brazil, with scarce resources.

However, the study has several flaws:

  • While the cross-sectional design is appropriate, there is no need to calculate a sample size for this kind of study. Indeed, you did not provide an intervention that you wanted to test. Oral clinical care should be standard care for PWHIV even if I understand that it may difficult to provide it in you area.  Therefore, your explanations on the sample size are wrong. What we want to know is how many patients you care for at the time of your study, and how many participated. Your study timeline is not mentionned: when did you provide this oro-dental assessment? If it is cross-sectional, it does not make sense to provide figures for the period running from March 1999 to March 2013.
  • Clinical assessment: how did you make the diagnosis of oral lesions? Was it a presumptive clinical assessment or did you perform swabs or even biopsy for definitive diagnosis? Your clinical methodology is not described. In particular, if only clinical assessment, I wonder how you made the difference between Herpes simplex and Herpes Zoster as it is quite difficult to differentiate herpes viruses clinically. You may have a list of clinical criteria to do so, and if it is the case, it must be described in your methods section.
  • Demographics characteristics are not all relevant or are missing.  Marital status may not influence the presence of oral lesions. Tabacco use et alcohol consumption, oral sex vs genital sex are missing and are easy to collect. 
  • Results according to viral load level and CD4 cells count: your findings are not a surprise as it is well documented since the beginning of the pandemic. You have an important number of unsuppressed patients (almost 70%). To characterize better your local epidemiology, it would have been interesting to know the time since ART initiation: are they newly started on treatment or failing patients? There is no information on patients with a CD4 between 50 and 200 in your description. Figure 2 is unnecessary. Prevalence of oral lesions according to viral load levels and CD4 count categories would have been informative.
  • Discussion: most of your discussion reports results of previous studies and should normally be part of an introduction section, not a discussion section.

Author Response

Reply to reviewer #4

1. Concern of the reviewer             

• While the cross-sectional design is appropriate, there is no need to calculate a sample size for this kind of study. Indeed, you did not provide an intervention that you wanted to test. Oral clinical care should be standard care for PWHIV even if I understand that it may difficult to provide it in you area.  Therefore, your explanations on the sample size are wrong. What we want to know is how many patients you care for at the time of your study, and how many participated. Your study timeline is not mentionned: when did you provide this oro-dental assessment? If it is cross-sectional, it does not make sense to provide figures for the period running from March 1999 to March 2013. 

Our response: Dear Reviewer #4, we appreciate your concern. The sample size was calculated at the beginning of the study, so by the time we designed the study, we authors did not know how many patients were using the dental service at Cacoal, therefore it make sense to calculate a sample size which later we added to the main text. Although we removed as requested.  

2. Concern of the reviewer             

• Clinical assessment: how did you make the diagnosis of oral lesions? Was it a presumptive clinical assessment or did you perform swabs or even biopsy for definitive diagnosis? Your clinical methodology is not described. In particular, if only clinical assessment, I wonder how you made the difference between Herpes simplex and Herpes Zoster as it is quite difficult to differentiate herpes viruses clinically. You may have a list of clinical criteria to do so, and if it is the case, it must be described in your methods section. 

Our response: Dear Reviewer #4, we appreciate your concern. The diagnosis method is mentioned at page 3, lines from 133 to 142 and the references which all clinical parameters used in our study were cited at page 3, lines from 133 to 142. I hope to clarify your concern. 

3. Concern of the reviewer             

• Demographics characteristics are not all relevant or are missing.  Marital status may not influence the presence of oral lesions. Tabacco use et alcohol consumption, oral sex vs genital sex are missing and are easy to collect.  

Our response: Dear Reviewer #4, we appreciate your suggestion. The text was carefully added to table 1. 

4. Concern of the reviewer             

• Results according to viral load level and CD4 cells count: your findings are not a surprise as it is well documented since the beginning of the pandemic. You have an important number of unsuppressed patients (almost 70%). To characterize better your local epidemiology, it would have been interesting to know the time since ART initiation: are they newly started on treatment or failing patients? There is no information on patients with a CD4 between 50 and 200 in your description. Figure 2 is unnecessary. Prevalence of oral lesions according to viral load levels and CD4 count categories would have been informative. 

Our response: Dear Reviewer #4, we appreciate your suggestion. Although it is a great ideia to identify the ART use time from all patients analyzed in this study, however this information is not available to all patients so we choose not to disclose the information. Regarding CD4 between 50 and 200 we really do not have this information at pronto to provide to readers. 

5. Concern of the reviewer             

• Discussion: most of your discussion reports results of previous studies and should normally be part of an introduction section, not a discussion section. 

Our response: Dear Reviewer #4, we appreciate your suggestion. The text was carefully added. 

Revised text: Page 8, lines 342-353,Another relevant point that might directly influenced the OL prevalence in PLWH in this study is the regular use of antiretroviral therapy (ART), since the advent of ART, the life expectancy of PLWH increased, as well as the access to information through internet, so the demand for dental treatment, including oral rehabilitation with dental implants, to improve dental quality of life increased significantly in the last few years [23-25]. The attempt to rehabilitate PLWH with dental implants has been reported since 1998 [26], since then papers seems to focus on healing, osseointegration process, longevity of dental implants, guided bone regeneration and peri-implant diseases. In most studies, the success of dental implant therapy in PLWH can be observed in the most diverse clinical situations and commonly in this study we found that the use of ART and medical control of comorbidities will directly influence the success of rehabilitation and the quality of life of the patient [27,28].” 

Round 2

Reviewer 3 Report

I reviewed the revised manuscript entitled “Evaluation of socio-demographic factors and prevalence of oral 2 lesions in people living with HIV from Cacoal, Rondônia, Am- 3 azon Region of Brazil”. It is necessary to revise the manuscript as follows:

Because most of the results of the paper are related to the socio-economic factors in people living with HIV and the prevalence of oral lesions is a small percentage of the results, the abstract, methodology, and discussion focused more on oral lesions prevalence. Therefore, according to the title, the manuscript should be revised by focusing on socio-economic factors in these patients.

Author Response

1. Concern of the reviewer

Because most of the results of the paper are related to the socio-economic factors in people living with HIV and the prevalence of oral lesions is a small percentage of the results, the abstract, methodology, and discussion focused more on oral lesions prevalence. Therefore, according to the title, the manuscript should be revised by focusing on socio-economic factors in these patients.   Our response:
Dear Reviewer #3, we appreciate your suggestion. Your idea is very interesting for the study, however this suggestion is opposing to the main objective of this study, which is to determine the prevalence of oral lesions and a possible correlation with lymphocytic indices and viral load, therefore developing the text focusing on other factors such as socioeconomic factors would need to change the main objective and all study text which would likely depart from our main objective. We understand that the occurrence of low prevalence is discouraging when we focus on the main objective of the study, but even if the low prevalence the result is still relevant to the scientific community, especially since it points out that the proper use of ART can help to reduce the risk of oral lesions from HIV.
